# Manipulating GA-Related Genes for Cereal Crop Improvement

**DOI:** 10.3390/ijms232214046

**Published:** 2022-11-14

**Authors:** Jingye Cheng, Camilla Beate Hill, Sergey Shabala, Chengdao Li, Meixue Zhou

**Affiliations:** 1Western Crop Genetics Alliance, College of Science, Health, Engineering and Education, Murdoch University, Perth, WA 6150, Australia; 2Tasmanian Institute of Agriculture, University of Tasmania, Hobart, TAS 7005, Australia

**Keywords:** gibberellin (GA), GA metabolism, GA signalling, plant growth, genetic modification, CRISPR/Cas9

## Abstract

The global population is projected to experience a rapid increase in the future, which poses a challenge to global food sustainability. The “Green Revolution” beginning in the 1960s allowed grain yield to reach two billion tons in 2000 due to the introduction of semi-dwarfing genes in cereal crops. Semi-dwarfing genes reduce the gibberellin (GA) signal, leading to short plant stature, which improves the lodging resistance and harvest index under modern fertilization practices. Here, we reviewed the literature on the function of GA in plant growth and development, and the role of GA-related genes in controlling key agronomic traits that contribute to grain yield in cereal crops. We showed that: (1) GA is a significant phytohormone in regulating plant development and reproduction; (2) GA metabolism and GA signalling pathways are two key components in GA-regulated plant growth; (3) GA interacts with other phytohormones manipulating plant development and reproduction; and (4) targeting GA signalling pathways is an effective genetic solution to improve agronomic traits in cereal crops. We suggest that the modification of GA-related genes and the identification of novel alleles without a negative impact on yield and adaptation are significant in cereal crop breeding for plant architecture improvement. We observed that an increasing number of GA-related genes and their mutants have been functionally validated, but only a limited number of GA-related genes have been genetically modified through conventional breeding tools and are widely used in crop breeding successfully. New genome editing technologies, such as the CRISPR/Cas9 system, hold the promise of validating the effectiveness of GA-related genes in crop development and opening a new venue for efficient and accelerated crop breeding.

## 1. Introduction

The Earth’s population is expected to reach 9.7 billion by 2050 [1]. To match this predicted population growth, annual food production should be doubled [2]. This task is challenged by the impact of climate change and associated abiotic stresses on agricultural production systems, as most of these people are living in Africa and South-eastern Asia—the regions most affected by climate constraints. Given the increasing rate of urbanisation and lack of available agricultural land, the only way to achieve this goal is to adopt genetic strategies to enhance crop productivity.

A major breakthrough in modern agriculture occurred in the 1960s, at the time of the “Green Revolution” [3], which embraced fertilization technologies that allowed a massive increase in the productivity of the land. One of the major contributors to the success of the “Green Revolution” was the introduction of high-yielding semi-dwarf crop varieties in rice and wheat. Old crop varieties/landraces of these species were characterised by tall, leafy, and weak stems, with a low harvest index of 0.3 (30% grain, 70% straw) [4]. Under the increased application of nitrogen fertilizers, these varieties were highly susceptible to lodging. The introduction of semi-dwarf varieties allowed an increase in crop yield due to increased lodging resistance [4,5].

Since then, many genes responsible for the semi-dwarf phenotype have been identified in major cereal crops such as wheat, barley, and rice. In rice, *sd-1*, encoding a defective gibberellin 20-oxidase (OsGA20ox), results in lower bioactive GAs, leading to a semi-dwarf phenotype with improved lodging resistance and higher crop production [6,7]. The barley 7-bp deletion mutant of *Sdw1/denso* that contains gibberellin 20-oxidase2 (*HvGA20ox*), the orthologous of *OsGA20ox2*, and the mutant forms of *REDUCED HEIGHT* (*TaRHT*) with a DELLA protein in wheat show a similar phenotype to rice *sd-1* [8,9,10].

The above “Green Revolution” genes are still being deployed in the breeding of modern crop varieties, all of which are involved in gibberellic acid (GA) biosynthesis pathways or GA response [7,9,11]. GA is an important phytohormone that regulates not only plant height but also key developmental events such as seed germination and flowering time [12]. Genes involved in GA biosynthesis and GA signalling pathways also have the potential to enhance crop yield. GA can also interact with other hormones to regulate plant development in response to stress. Thus, the role of GA goes well beyond its impact on plant height and may impact crop yield via numerous pathways. The aim of this review is to (1) discuss the role of gibberellin in plant growth and development; (2) explore its interaction with other phytohormones in manipulating plant development; and (3) summarise the current knowledge of the genetic basis for improving agronomic traits by targeting GA signalling pathways.

## 2. Gibberellin Biosynthesis and Deactivation Pathways

More than 130 GA have been reported in plants, fungi, and bacteria to date [13]. They are mostly biologically inactive and act as precursors for bioactive GAs, including GA1, GA3, GA4, and GA7 [14]. GAs are derived from a common diterpenoid precursor, which requires three different oxidative enzymes for biosynthesis: terpene synthases (TPSs), cytochrome P450 mono-oxygenases (P450s), and 2-oxoglutarate-dependent dioxygenases (2ODDs) [15]. In higher plants, GAs are formed mainly in the methylerythritol phosphate pathway [16], by which the hydrocarbon intermediate *ent-kaurene* is produced from a separate pool of *trans-geranylgeranyl diphosphate* (GGPP) via a GGPS (GGPP synthase) in proplastids [17]. The two-step synthesis of *ent-kaurene* from GGPP via *ent*-copaly1 is catalyzed by separating bifunctional enzymes encoded by the *TPS* genes *ent*-copaly1 diphosphate synthase (CPS) and *ent*-kaurene synthase (KS). CPS is a class II (proton-initiated) cyclase, and KS is a class I (initiated by phosphate ionisation) cyclase (Figure 1).

In Arabidopsis, both CPS and KS are encoded by single genes, which, when inactivated, cause the phenotypes of severe GA deficiency, characterised by extreme dwarfism, male and female infertility, and non-germinating seeds [18]. In rice, only single members of each gene family in both CPS- and KS-like genes are involved in GA biosynthesis, while others are dedicated to phytoalexin production [19,20,21].

The transformation of *ent*-kaurene into bioactive forms involves two oxidative enzymes, namely, cytochrome P450 mono-oxygenases (P450s) and 2-oxoglutarate-dependent dioxygenases (2ODDs) [15]. KO (*ent*-kaurene oxidase) and KAO (*ent*-kaurenoic acid oxidase) are two mono-oxygenases, which are required to catalyse the formation of GA12, a common precursor for all GAs in plants [22]. KO and KAO belong to the CYP701A P450 clade and CYP88A clade, respectively [23]. Both enzymes are biologically active in the endoplasmic reticulum, whereas KO is also localised in the plastid envelope [24]. The oxidation of *ent*-kaurene to *ent*-kaurenoic acid by KO occurs in three steps via the intermediates *ent*-kaurenol and *ent*-kaurenal, depending on the mechanism for hydroxylations on C-19 [25]. In rice, a cluster of five KO-like genes was found on chromosome 6 and the mutation site of *osko2-1*, which presents a severe GA deficiency and a dwarf phenotype without flowering [20]. Thus, only one gene of the cluster, *OsKO2* (*CYP701A6*), is required for GA biosynthesis [20,26].

The next three steps of KAO-catalysed oxidation of *ent*-Kaurenoic acid to GA12 occur via *ent*-7α-hydroxykaurenoic acid and GA12-aldehyde, requiring successive oxidations at C-7β, C-6β, and C-7 [27]. Studies show that rice only contains a single KAO-like gene, whereas KAO is encoded redundantly by two genes in *Arabidopsis thaliana*, *Pisum sativum*, and *Helianthus annuus* L. [28,29].

GA12, a product of KAO, lies at a key point in the pathway in higher plants and is hydroxylated at C-13 and/or C-20 [17]. GA12 can be hydroxylated by gibberellin 13-oxidase (GA13ox) to form GA53 [30]_._ The next steps occur in the cytoplasm where enzymes belonging to the gibberellin 20-oxidase (GA20ox) family promote the final conversion of GA12 and GA53 in parallel pathways into bioactive GA9 and GA20, respectively [17]. This is a three- or four-step process, which involves the repeated loss of C-20 [31]. Previous studies indicate the conversion of 20-methyl compounds to C19-lactone via 20-alcohol and 20-aldehyde [32]. The C-20 alcohol lactonises with the C-19 carboxylic acid group in acidic conditions to inhibit further oxidation by GA20ox. However, C-20 lactone can be oxidised by a 20-oxidase activity in some tissues, including the roots, leaves, and shoot apical meristem.

In the final step, to produce biologically active hormones, the conversions of GA9 and GA20 to GA4 and GA1, respectively, involve hydroxylations on C-3β catalysed by the ODD gibberellin 3-oxidase (GA3ox) [33]. Moreover, GA9 is also transformed into GA7 via 2,3-didehydro-GA9 [34]. Similarly, in dicots, GA20 is also transformed into GA3, catalysed by a single GA3ox via GA5 as the intermediate, in addition to being transformed into GA1 [35].

Several mechanisms have been detected to regulate GA homeostasis by inactivating GAs [36], with 2β-hydroxylation being predominant. GA2oxs (GA 2-oxidases) and ODDs are enzymes involved in this activity. On the basis of their function, GA2oxs can be divided into two major groups: one acting on C19-GAs, whereas the other acts on C20-GAs [37]. There are two subgroups of C19-GA2oxs. These two groups differ not only in amino acid sequence but in biochemical function, as proposed recently [38,39]. The first subgroup is bifunctional, acting as 2β-hydroxylase to oxidise the second subgroup to a ketone; the second subgroup is unable to produce GA catabolites, a class of isolated 2-keto products resulting from further oxidisation of the 2β-hydroxy group by the first subgroup [39]. The C20-GAs are considered a separate family from C19-GAs [40].

Other mechanisms for GA inactivation mainly include epoxidation via GA methyl transferase 1 (GAMT1) and GA methyl transferase 2 (GAMT2). GAMT1 and GAMT2 are two members of the SABATH group of methyl transferases found in Arabidopsis and are specific for GAs. The *GAMT* genes are expressed in developing seeds and encode enzymes that methylate the 6-carboxyl group of C19-GAs, which leads to GA inactivity. Lines with *GAMT* gene knockout show an increased content of GAs, confirming their roles in GA regulation. It has been reported that the product of a rice *EUI (ELONGATED UPPERMOST INTERNODE)* gene, a cytochrome P450 mono-oxygenase (CYP714D1), can convert GAs into their 16α, 17-epoxides [41]. Epoxidation results in GA deactivation, as confirmed by the overexpression of *EUI* in rice, which causes dwarfism and reduced GA4 content in the uppermost internode. This effectivity of epoxidation varies with the specific substrates of GAs, with GA12, GA9, and GA4 being more effective.

Furthermore, some transcription factors are involved in the upstream regulation of GA biosynthesis in plants. For example, in Arabidopsis, the overexpression of *WUSCHEL-RELATED HOMEOBOX 14* (*WOX14*) promotes GA biosynthesis by stimulating *GA3ox* gene expression and represses *GA2ox* genes [42]. The transcriptional factor helix–loop–helix (bHLH) proteins are also involved in GA biosynthesis regulation. INDEHISCENT (IND) promotes GA biosynthesis via directly activating *GA3ox1* [43], while ALCATRAZ (ALC)/Phytochrome (phy)-Interacting Factor 3 (PIF3)/Phytochrome (phy)-Interacting Factor 4 (PIF4) regulate GA signalling by interacting with DELLA [43,44,45]. Meanwhile, in tomatoes, the overexpression of *SlbHLH95* downregulates the two GA biosynthesis genes *SlGA20ox2* and *SlKS5*, leading to decreased GA biosynthesis [46].

## 3. GA Signalling Pathway and Network

GA acts as an important phytohormone that plays a major role in most crop developmental processes, including seed germination, stem elongation, leaf expansion, and flowering, by increasing cell division and elongation (Figure 2) [47,48,49,50].

### 3.1. GA Signalling Pathway

Bioactive GAs activate GA signalling, which is one of the major pathways that mediates plant development [15,51]. GA signalling is largely regulated by DELLAs, which are a class of nuclear proteins [52]. DELLAs act as a negative regulator of GA signalling and can be degraded through the ubiquitin–proteasome system [53,54]. GAs promote a conformational change in the nuclear receptor GA-insensitive dwarf1 (GID1) that enhances the interaction between GID1 and DELLA proteins [45,53,55]. This interaction stimulates the binding of the E3 ubiquitin Ligase SLEEPY1 (SLY1) to DELLA, triggering its degradation by the 26S proteasome pathway in the nucleus [56,57]. After DELLAs target 26S proteasome-mediated proteolysis, downstream genes start to express, leading to GA-dependent responses that regulate the plant developmental process [58]. The proteolysis of DELLA can lead to the upregulation of four target genes, including *PHYTOCHROME-INTERACTING FACTORs* (*PIFs*), *GAMYB*, *SUPPRESSOR OF CONSTANS 1 (SOC1)*, and *SQUAMOSA PROMOTER-BINDING PROTEIN-LIKEs* (*SPLs*), and the downregulation of *SPINDLY* (*SPY*). DELLA-mediated *PIFs* are required for coordinating light and GA signals to regulate hypocotyl elongation [59]. *GAMYB* is the downstream gene of DELLA and positively regulates GA-responsive genes that are involved in mediating flowering. For example, one of the *GAMYB*-like genes, *MYB33*, directly activates the floral meristem identity gene *LEAFY* (*LFY*) to promote flowering [60]. The upregulation of SPLs activates the *microRNA172–FLOWERING LOCUS T* (miR172-FT) module, leading to increased *APETALA1* (*AP1*) and upregulated SOC1, which are two other pathways involved in GA signalling to activate *LFY* for promoting flowering [61,62,63,64]. In addition to DELLA, *SPY* encoding nucleocytoplasmic protein is another negative regulator of GA signalling by activating the GlcNAcylation of DELLA [65]. The downregulation of *SPY* could partially rescue the non-germination and dwarf phenotype caused by GA-deficient mutants [66,67].

### 3.2. Crosstalk between GA and Other Plant Hormones in Plants

GA not only affects phenology development via the endogenous and environmental exogenous genetic pathways but also interacts with phytohormone pathways, such as abscisic acid (ABA), jasmonic acid (JA), cytokinins (CK), auxin (IAA), and brassinosteroids (BR), to control plant phenology.

#### 3.2.1. Antagonistic Regulation of Phytohormones with GA in Mediating Plant Growth

In Arabidopsis, ABA antagonises GA in several developmental processes, especially the regulation of seed dormancy and germination. ABA is synthesised by 9-cis-epoxycarotenoid dioxygenase (NCED) and activates ABA-INSENSITIVE (ABI) to regulate the balance between ABA and GA. The interaction between ABI3 and ABI5 regulates ABA and GA metabolic genes by activating SOMNUS (SOM) to regulate seed germination at high temperatures [68]. In addition to temperature, ABI5 is a key mediator in light–ABA/GA networks during seed germination [69]. ABI5 can also be activated by ABI4 and, subsequently, induces a DELLA gene, *RGA*-*like2* (*RGL2*), by binding to its promoter region. DELLA activates *XERICO* to increase ABA biosynthesis [70]. ABI4, an APETALA2 (AP2)-domain-containing transcription factor (ATF), plays a central role in mediating ABA/GA homeostasis and antagonistic regulation in the post-germination stage and flowering period by activating NCED6 and a GA2oxs, such as GA2ox7 [71,72]. In addition, a recent study showed that GA12 16, 17-dihydro-16α-ol (DHGA12) produced by gain-of-function in ABA-modulated Seed Germination 2 (GAS2) could bind to GA receptor GID1c to promote seed germination, hypocotyl elongation, and cotyledon greening by altering the ABA/GA ratio [73]. In rice, *OsAP2-39* (includes an APETALA 2 (AP2) domain) operates as a central regulator in the antagonistic network between ABA and GA that controls plant growth. *OsAP2-39* upregulates *OsNCED-1*, which leads to increased ABA and the elongation of the uppermost internode (EUI), which are involved in the deactivation of GAs [74].

In addition, excessive GA activates ABA catabolism, leading to ABA reduction and, thus, accelerates flowering and spikelet initiation at the central part of the barley spike [75,76,77]. The crosstalk between GA and ABA also regulates biotic stress. For example, ABA and GA have the opposite effects on Fusarium head blight (FHB) infection, which is a devasting disease caused by *Fusarium graminearum* [78,79].

JA, normally induced by abiotic stress, antagonises the GA response through the interaction between the JASMONATE-ZIM domain (JAZ) and DELLA, which regulates the balance between plant defence and growth [80]. When JA signalling is downregulated, accumulated JAZ targets DELLA, and PIFs are activated for enhancing the GA response to regulate plant growth. However, with upregulation of JA signalling, upon insect or pathogen attack, JAZ repressors are degraded, and the accumulated DELLA inhibits the transcription of PIFs, leading to slow growth. At the same time, the downstream gene Myelocytomatosis (MYC) involved in the JA signalling branch is accumulated, leading to plant defence [81,82]. Only two of twelve *OsJAZs*, *OsJAZ8* and *OsJAZ9*, interact with SLRs in rice to regulate the antagonistic mechanism of JA and GA [82].

CK is another phytohormone that has an antagonistic effect on GA in a wide variety of developmental activities, including cell differentiation, meristem maintenance, as well as shoot and root elongation. KNOXs act as the key intermediate regulators between GA and CK levels in the shoot apical meristem (SAM), which activate *GA2ox* and promote the expression of *ISOPENTENYL TRANSFERASE7 (IPT7)*, a CK biosynthesis gene [83,84,85]. The crosstalk between CKs and GAs plays a significant role in mediating flower initiation and increased floral productivity [86,87].

Similar to Arabidopsis, five functional class I KNOX genes in rice have a similar function that positively regulates CK signalling by activating IPT [88]. OsIPT2 and OsIPT3, two of eight IPTs in rice upregulated by KNOX, lead to a decreased GA content and increased CK required for the initiation of SAM [89,90].

#### 3.2.2. Synergetic Regulation of Phytohormones with GA in Mediating Plant Growth

Auxin regulates GA signalling and biosynthesis in mediating plant growth. The major bioactive auxin is indole-3-acetic acid (IAA), which is rapidly synthesised in the tissues with extended cell division [91]. The transcription factors of the *AUXIN RESPONSE FACTOR* (*ARF*) family, as well as AUXIN INDUCIBLE/INDOLE-3-ACETIC ACID INDUCIBLE (AUX/IAA), are involved in auxin response. These not only regulate GA metabolism enzymes, including GA20ox, GA3ox, and GA2ox, but also negatively regulate DELLA [92,93]. Most studies have revealed that the crosstalk between GA and auxin affects the root, hypocotyl, uppermost internode (UI), and floral transition. For example, DELLA proteins sequester ARFs to block xylem expansion in the second phase of hypocotyl before flowering, while after flowering, GA accumulates in hypocotyl to degrade DELLA, releasing ARFs (ARF6-8) to promote cambium senescence, phloem repression, and fiber differentiation [94]. In rice, the decreased panicle-derived IAA results in GA1 deficiency by downregulating *OsGA3ox2* expression levels, leading to decreased cell numbers and cell elongation, thereby resulting in a shortened UI [95]. Another finding showed that a decreased level of IAA/GA1 leads to low spikelet fertility under high temperatures [96]. IAA positively regulates GA biosynthesis for mediating the development of the apical part of the spike in barley [77].

BR is considered to act synergistically with GA in mediating the numerous aspects of plant growth. BR induces the inactivation of BRASSINOSTEROID-INSENSITIVE 2 (BIN2), which allows BRASSINAZOLE-RESISTANT1/BRI1-EMS-SUPPRESSOR 1 (BZR1/BES1) to be rapidly dephosphorylated and activated by PHOSPHATASE 2A (PP2A) and translocated into the nucleus to activate PRE1 and repress IBH1. An antagonistic cascade formed by PRE1, IBH1, and HBI1 regulates plant growth [97]. GAs crosstalk with BR to control cell elongation and plant growth through DELLAs and BZR1/BES1 [98,99]. The physical interaction between DELLAs and BZR1 and BES1 inhibits theirtranscriptional activities on downstream target genes [98,99]. BR interacts not only with GA signalling but also the feedback response of GA biosynthesis. The regulatory role of DELLAs in BES1 regulation also activates *GA20ox1* and *GA3ox1* [98,100,101].

BR also promotes cell elongation by regulating GA metabolism genes in rice. For example, BR significantly induces D18/GA3ox2, resulting in increased levels of GA1 in rice seedlings. Compared to Arabidopsis, rice has a divergent pattern, wherein the responding pathway is highly dependent on tissue and endogenous hormone levels. For instance, the excessive application of bioactive BR induces GA2ox3 transcriptional activity to induce GA inactivation. In the feedback mechanism, leaf angles enlarged by the enhanced BR can be reduced by GA treatment [102]. Moreover, two other pathways involved in regulating leaf angles based on the interaction of BR and GA, the GA–GID1–DELLA pathway and the G protein-dependent pathway in rice have been characterised [103,104,105] (see Figure 2 below).

## 4. GA-Related Genes Regulate Agronomic Traits

### 4.1. Seed Dormancy and Germination

Seed dormancy is an essential agronomic trait that allows the crop to germinate under favorable conditions but delays this process when seeds are still located on the mother plant. GA plays a significant role in the transition of the seed from dormancy to germination. Several genes acting as GA response regulators that promote seed germination have been identified in cereal crops (Table 1) [106,107].

In Arabidopsis, *CYTOCHROME P450* genes encode gibberellin 13-oxidase, which plays a significant role in manipulating the development of seeds/siliques. *cyp72a9* mutants exhibit accelerated germination due to an increased level of GA4 [108]. Besides the genes involved in the GA biosynthesis pathway, *RGL2*, one of three *RGA*-*like* (*RGL*) genes involved in the DELLA subfamily, is the major negative regulator of seed germination [109], with *rgl2* alleles being more resistant to the inhibitory effect of paclobutrazol (PAC) on germination than *rgl1-1*, *gai-t6*, and *rga-t2*. Loss-of-function mutations in *RGL2* completely restore germination, in the absence of exogenous GA, to *ga1-3* seeds with a nongerminating phenotype [110]. In addition to the DELLA-dependent pathway, *SPINDLY* (*SPY*) is also involved in GA signalling to control seed germination, as SPY can partially regulate the GA signal transduction pathway that is independent of GA. *AtSPY* is considered the negative regulator of GA signalling, and *SPY* mutants promote seed germination by conferring resistance to PAC with an inhibitory effect on seed germination and rescue the non-germinating phenotype in *ga1-2*, an allele of *ga1-3* [67,111].

The *SPY* homologues in cereal crops also have a similar function. For example, *HvSPY* can suppress the expression of α-amylase induced by GA and maintain seed dormancy [130,138]. However, there are unique genes in cereals with dominant functions in seed germination. For instance, *SLENDER1* (*HvSLN1*) is another repressor of α-amylase activity induced by GA in barley seeds, and the mutation in *HvSLN1* exhibits non-dormancy with high α-amylase expression in the aleurone [131,132,139,140]. In rice, the mutation of *LEA33*, one of five *Late Embryogenesis Abundant* (*LEA*) genes, promotes seed germination via enhancing GA biosynthesis in rice embryos [116]. GA oxidases compose another big family involved in GA synthesis. It was shown that the upregulation of *TaGA20ox1* and *TaGA3ox2* in wheat led to breaking seed dormancy through increasing GA content [35,125].

Thus, current studies are in agreement that DELLA may be the most direct regulator of GA-regulated agronomic traits and abiotic stress tolerance. Moreover, orthologous genes generally have the same biochemical function and exhibit a similar phenotype in cereal crops. The genes involved in GA metabolism and GA signalling pathways for regulating seed dormancy and germination are either DELLA-dependent or -independent GA responses. Meanwhile, several DELLA proteins that regulate seed germination in Arabidopsis are orthologous to GA response height-mediating factors in cereal crops such as RGL1/RGL2 [110]. DELLA proteins, REPRESSOR OF GA1-3/GA INSENSITIVE (RGA/GAI), in Arabidopsis and their orthologous genes in cereal crops inhibit stem elongation [110,117,126,132,133,141], and the upregulating DELLA protein SLENDER1 (SLR1) in rice can also enhance submergence [122].

### 4.2. Plant Stature

GA is also a crucial factor that determines plant stature such as plant height and biomass. Plant height is an important agronomic trait that plays a central role in crop performance. Plant height is positively correlated with its biomass during the vegetative stage, while short plants possess higher resistance to lodging. Therefore, by modifying GA-related genes, plant height and yield components can be manipulated. 

In Arabidopsis, *RGA* [112] and *GAI* [113], which share a similar amino acid sequence with *RGL2*, are considered negative regulators of the GA response and inhibit stem elongation [111]. The functional orthologous gene of *RGA*/*GAI* in rice is *OsSLR1*, a DELLA protein, which is a repressor of GA signalling [117]. The overexpression of *OsSLR1* reduces GA responsiveness and exhibits dwarf phenotype in rice [126,141].

*HvSLN* in barley and *REDUCED HEIGHT* (*TaRHT*) in wheat are found to be the orthologous genes of *OsSLR.* Both genes have an effect, similar to *OsSLR*, on plant height [132,133].

The well-known “Green Revolution gene” in rice has been mapped as *OsGA20ox2* at the *SD1* locus. The removal of this locus causes a semi-dwarf phenotype due to the low amount of active GA in the shoot [118]. Moreover, the mutation in *OsGA20ox3* exhibits a similar semi-dwarf phenotype, while the overexpression of *OsGA20ox3* shows an elongation phenotype [123]. Similarly, *OsGA2-oxidase 6* (*OsGA2ox6*) is related not only to the semi-dwarf phenotype but also more tillers, leading to a high yield [119]. In barley, *HvGA20ox2*, located in semi-dwarf locus *sdw1/denso*, is homologous to *OsGA20ox2* [134] and has a similar function.

A few years ago, *Rht24* in wheat was identified as a novel locus for height reduction, which encodes a GA2 oxidase [142]. Interestingly, the manipulation of *TaGA2oxA9* can not only reduce plant height without yield loss but also increase plant nitrogen use efficiency [129].

However, GA-related semi-dwarfing genes can also have adverse effects on agronomic performance such as decreased coleoptile length [143,144,145]. Coleoptile length is essential for deep sowing in dry land agriculture regions [146,147]. Some alleles of semi-dwarfing genes can also have negative impacts on yield components. For example, Kandemir et al. found that one of the *sdw1/denso* alleles, *sdw1.d* with a 7 bp deletion in exon1, reduces the thousand-seed weight [148].

### 4.3. Abiotic Stress Tolerance

Genes encoding dioxygenases, including *GA20ox*, *GA3ox*, and *GA2ox*, act as the main regulators in GA biosynthesis for environmental signals; of particular interest is the role of *GA2ox* genes in response to abiotic stress. In Arabidopsis, *DWARF AND DELAYED FLOWERING 1* (*DDF1*) interacts with *AtGA2-oxidase 7* (*AtGA2ox7*) to upregulate gene expression, which leads to GA deficiency and enhanced salt tolerance [114]. In rice, the ectopic expression of *OsGA2ox6* and *OsGA2-oxidase 8* (*OsGA2ox8*) promotes the accumulation of osmoregulators such as proline, and, thus, plants can adapt to osmotic stress and drought [119,120].

GA also plays a central role in mediating crop shoot elongation to improve crop survival under waterlogging conditions [149]. Under waterlogging, the Green Revolution gene *SD1* activated by *ETHYLENE-INSENSITIVE 3-like 1a* (*OsEIL1a)*, an ethylene-responsive transcription factor, promotes GA synthesis, particularly GA4, leading to stem elongation above the water surface to restore gas exchange between crop tissues and air [121]. *Submergence 1A* (*Sub1A*) in rice acts as an ethylene response factor that confers submergence tolerance by impeding shoot elongation via stimulating the GA signalling repressors *SLR1* and *SLR1 Like-1* (*SLRL1*) [122].

In wheat and barley, GA-related genes, such as *Rht12*, *RhtB1b*, and *sdw1*, can enhance heat and drought stress tolerance through an expanded growth period, increased tiller number, improved lodging resistance, and preventing head loss, thus, improving crop adaptability and grain quality [127,128,135,136].

### 4.4. Biotic Stress Tolerance

Biotic stress, including bacteria, fungi, viruses, nematodes, insects, and infections, is another major environmental factor that affects crop productivity [150].

*AtGA2ox7* in Arabidopsis is involved in enhancing plant resistance to pathogens [115]. In rice, *OsGA20ox3* RNAi lines show improved resistance to rice blast and pathogens X. oryzae pv. Oryzae and upregulated defence-related genes, while plants with an overexpression of *OsGA20ox3* are more sensitive to pathogens [123]. A GA-insensitive severe dwarf mutant, *gibberellin-insensitive dwarf1* (*Osgid1*), and *probenazole-inducible protein* (*PBZ1*), regulated by GA signalling, are involved in resistance to rice blast fungus [124]. Genes involved in GA regulating pathways that are associated with resistance to biotic stress in wheat and barley remain mostly unexplored.

## 5. CRISPR/Cas9-Mediated Gene Editing of Gibberellin Genes for Crop Improvement

CRISPR/Cas9 technology has shown great promise for functional gene research and crop improvement and has been used to validate gene functions in the model plant *Arabidopsis thaliana*, as well as the crop species rice, wheat, barley, maize, tomato, and grape [151,152,153]. Genetic improvement studies have applied the CRISPR/Cas9 system to induce mutations in GA-related genes. For example, *ZmGA20ox3* in maize [154], *PROCERA* encoding DELLA protein in tomato [153], and *MaGA20ox2* in banana [155] were edited by CRISPR/Cas9 technology to create semi-dwarf phenotypes that help to increase lodging resistance and mechanical harvest. In rice, *OsGA3ox1* was also verified as a significant gene in regulating starch accumulation in mature pollen grains [156]. Moreover, the mutations of *HvARE1* edited by gene editing led to increased nitrogen use efficiency in barley [157].

## 6. Conclusions and Perspectives

In the past half-century, the use of *SD1* in rice and *sdw1/denso* in barley, both encoding GA20-oxidase, has dramatically increased grain yield and led to paradigm-shifting agricultural practices termed a “Green Revolution”. Many alleles of *sd1* and *sdw1/denso* with economic values were identified through fine-tuning plant phenology and yield potential. However, only a few of them have been wildly used in breeding programs on a global scale. In addition, the allelic effects on crop agronomic performance should be explored in diverse genetic backgrounds to validate the effectiveness of gene function and the adverse influence on other agronomic traits. Therefore, the modification of GA-related genes and the identification of novel alleles without a negative impact on yield and adaptation are significant in cereal crop breeding for plant architecture improvement.

With the arrival of the era of genome editing technologies (especially the CRISPR/Cas9 system), the functional analysis of genes involved in GA signalling and the investigation of functional alleles of known GA-related genes can be conducted in a more efficient and less time-consuming manner, when compared with the conventional breeding. The use of genome editing technology, such as CRISPR/Cas9, to modify GA-related genes can alter plant morphology and performance and, thus, generate a wild range of germplasms for specific environmental conditions and agricultural practices.

## Figures and Tables

**Figure 1 ijms-23-14046-f001:**
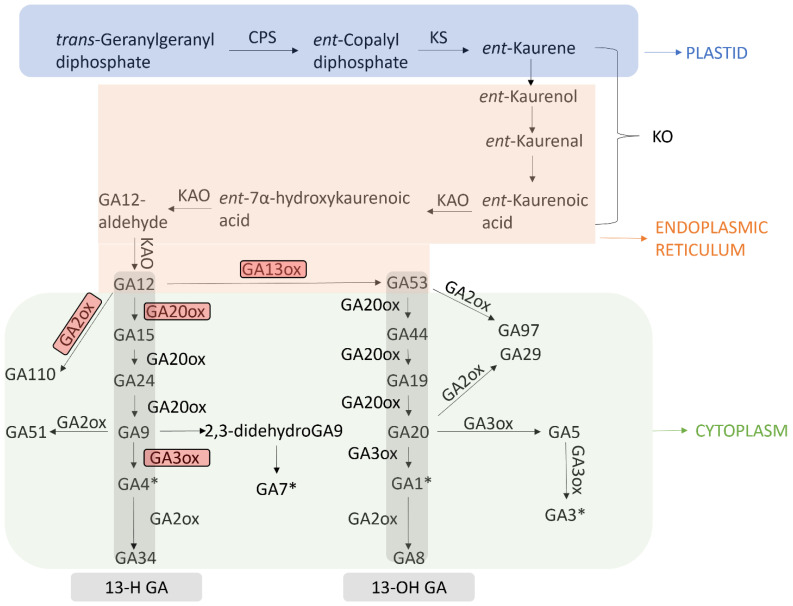
Gibberellin (GA) biosynthesis and deactivation pathways in Arabidopsis. CPS: *ent*-copaly1 diphosphate synthase; KS: *ent*-kaurene synthase; KO: *ent*-kaurene oxidase; KAO: *ent*-kaurenoic acid oxidase; GA13ox: GA 13-oxidase; GA20ox: GA 20-oxidase; GA3ox: GA 3-oxidase; GA2ox: GA 2-oxidase; red boxes indicate key enzymes; asterisks indicate bioactive GA; and 13-H GAs and 13-OH GAs are highlighted in grey; *: bioactive GA.

**Figure 2 ijms-23-14046-f002:**
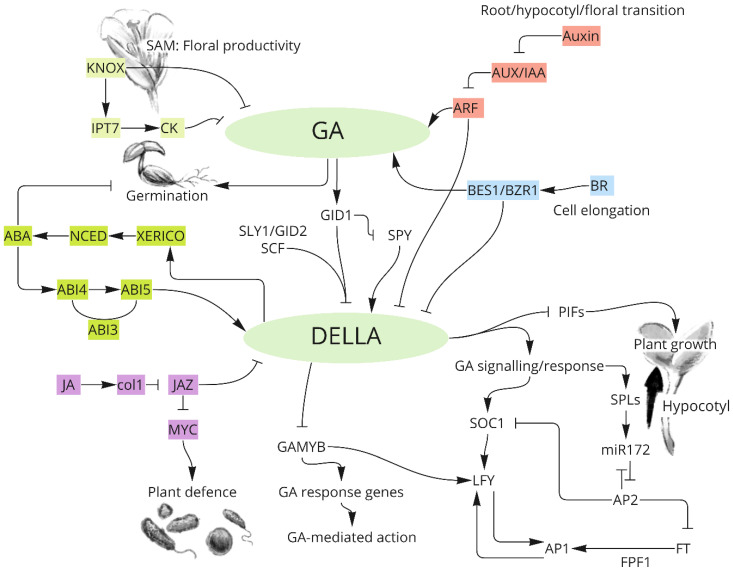
Gibberellin (GA) signal transduction pathways and crosstalk between GA and phytohormones in mediating key developmental events in Arabidopsis. KNOX: KNOTTED1-like homeobox; IPT7: ISOPENTENYL TRANSFERASE7; CK: cytokinin; NCED: 9-cis-epoxycarotenoid dioxygenase; ABA: abscisic acid; ABI3: ABA-INSENSITIVE 3; ABI4: ABA-INSENSITIVE 4; ABI5: ABA-INSENSITIVE 5; COI1: coronatine-insensitive 1; JAZ: JASMONATE-ZIM domain; MYC: Myelocytomatosis; JA: jasmonic acid; AUX/IAA: Indole-3-acetic Auxin/Acid Inducible; ARFs: AUXIN RESPONSE FACTORs; BR: brassinosteroids; BES1/BZR1: BRASSINAZOLE-RESISTANT1/BRI1-EMS-SUPPRESSOR 1; PIFs: PHYTOCHROME-INTERACTING FACTORs; SPLs: SQUAMOSA PROMOTER-BINDING PROTEIN-LIKEs miR172: microRNA172; AP2: APETALA2; FT: FLOWERING LOCUS T; AP1: APETALA1; FPF1: FLOWERING PROMOTING FACTOR 1; SOC1: SUPPRESSOR OF CONSTANS 1; and LFY: LEAFY. The arrows indicate stimulatory effects, and T sharp symbol indicates inhibitory effects.

**Table 1 ijms-23-14046-t001:** Genes involved in GA metabolism and signalling for regulating plant growth.

Gene Name (Abbreviation)	Plant Species	Annotation	Function	Reference
*CYTOCHROME P450, FAMILY 72, SUBFAMILY A, POLUPEPTIDE 9* (*CYP72A9*)	*Arabidopsis thaliana*	Encode gibberellin 13-oxidase	Negatively regulate bioactive GA4 to maintain seed dormancy.	[108]
*RGA*-*like2* (*RGL2*)	*Arabidopsis thaliana*	Encode a DELLA protein	Negatively regulate GA signalling response to decrease seed germination rates.	[109,110]
*SPINDLY* (*SPY*)	*Arabidopsis thaliana*	An O-linked N-acetylglucosamine (GlcNAc) transferase (OGT); negative regulator of the GA signaling pathway;	Negative regulator of GA biosynthesis pathway to inhibit seed germination frequency.	[67,111]
*REPRESSOR OF GA* (*RGA*)	*Arabidopsis thaliana*	Encode a DELLA protein; repressor of GA signalling pathway	Negatively regulate GA signalling response to negatively mediate stem elongation.	[112]
*GA INSENSITIVE* (*GAI*)	*Arabidopsis thaliana*	Encode a DELLA protein; a repressor of GA response	Negatively regulate GA signalling response to inhibit stem elongation.	[113]
*GA2-oxidase 7* (GA2ox7)	*Arabidopsis thaliana*	Encode a gibberellin catabolic enzyme gibberellin 2-oxidase that acts specifically on C-20 gibberellins	Reduce endogenous GA biosynthesis to improve salinity tolerance and resistance to pathogens.	[114,115]
*Late Embryogenesis Abundant 33* (*LEA33*)	*Oryza sativa*	Regulate *OsGA20ox1* to mediate gibberellin biosynthesis	Negative regulator of GA biosynthesis to negatively regulate grain size and seed germination rates.	[116]
*SLENDER RICE1* (*SLR1*)	*Oryza sativa*	Encode a DELLA protein; an O-linked N-acetylglucosamine transferase; negative regulator of GA response	Negatively regulate GA signalling pathway to inhibit stem elongation.	[117]
*GA20-oxidase2* (*GA20ox2*)	*Oryza sativa*	Encode a gibberellin biosynthetic enzyme gibberellin 20-oxidase	Enhance GA biosynthesis to positively regulate stem elongation.	[118]
*GA2-oxidase6* (*GA2ox6*)	*Oryza sativa*	Encode a gibberellin catabolic enzyme gibberellin 2-oxidase	Reduce endogenous GA biosynthesis to decrease plant height, increase tiller number, and modify crop architecture to enhance drought stress tolerance.	[119]
*GA2-oxidase8* (*GA2ox8*)	*Oryza sativa*	Encode a gibberellin catabolic enzyme gibberellin 2-oxidase	Positively regulate osmotic regulators and antioxidase to enhance osmotic stress tolerance.	[120]
*Semi-dwarf1* (*SD1*)	*Oryza sativa*	Encode a gibberellin biosynthetic enzyme gibberellin 20-oxidase	Enhance endogenous GA biosynthesis to promote stem elongation to escape waterlogging.	[121]
*Submergence 1A* (*Sub1A*)	*Oryza sativa*	An ethylene response factor; promote GA signalling repressors, *SLR1* and *SLRL1*, activities	Negatively regulate GA signalling to limit shoot elongation to escape submergence stress.	[122]
*GA20-oxidase3* (*GA20ox3*)	*Oryza sativa*	Encode a gibberellin biosynthetic enzyme gibberellin 20-oxidase	Knockout reduces the endogenous GA to inhibit stem elongation and improve resistance to pathogens.	[123]
*Gibberellin-insensitive dwarf1* (*gid1*)	*Oryza sativa*	A soluble gibberellic acid receptor	A recessive GA-insensitive severe dwarf mutant and positively regulate resistance to rice blast fungus.	[124]
*GA20-oxidase1* (*GA20ox1*)	*Triticum aestivum*	Encode a gibberellin biosynthesis enzyme gibberellin 20-oxidase	Enhance endogenous GA biosynthesis to positively regulate seed dormancy breakage and seed germination.	[35,125]
*GA3-oxidase2* (*GA3ox2*)	*Triticum aestivum*	Encode a gibberellin biosynthesis enzyme gibberellin 3-oxidase	Enhance endogenous GA biosynthesis to positively regulate seed dormancy breakage and seed germination.	[35,125]
*REDUCED HEIGHT-B1b/D1b* (*RhtB1b/D1b*)	*Triticum aestivum*	Encode a DELLA protein	Negatively regulate GA signalling response to reduce stem elongation and enhance resistance to high temperature and drought stress.	[126,127]
*REDUCED HEIGHT 12* (*Rht12*)	*Triticum aestivum*	Encode a DELLA protein	Negatively regulate GA signalling response to inhibit stem elongation and enhance resistance to high temperature and drought stress.	[128]
*GA2-oxidaseA9* (*GA2oxA9*)	*Triticum aestivum*	Encode a gibberellin catabolic enzyme gibberellin 2-oxidase	Reduce endogenous GA biosynthesis to negatively regulate stem elongation without yield loss.	[129]
*SPINDLY* (*SPY*)	*Hordeum vulgare*	A negative regulator of GA response	Negatively regulate GA signalling to suppress the expression of α-amylase induced by GA and maintain seed dormancy.	[130]
*SLENDER1* (*SLN1*)	*Hordeum vulgare*	Encode a DELLA protein; negatively regulate GA signalling	Negatively regulate GA signalling response to suppress the expression of α-amylase and maintain seed dormancy, as well as negatively regulate stem elongation.	[131,132,133]
*GA20-oxidase2/Sdw1/denso* (*GA20ox2*)	*Hordeum vulgare*	Encode a gibberellin biosynthetic enzyme gibberellin 20-oxidase	Enhance endogenous GA biosynthesis to positively regulate stem elongation and enhance resistance to drought stress.	[134,135,136]
*PANICLE RACHIS LENGTH 5* (*Prl5*)	*Oryza sativa*	Encode a gibberellin biosynthesis enzyme gibberellin 20-oxidase4	Positively regulate endogenous GA biosynthesis to increase panicle rachis elongation.	[137]

## Data Availability

Data Sharing not applicable.

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
