# Peer review of "Manipulating GA-Related Genes for Cereal Crop Improvement"

_ijms, 2022, doi:10.3390/ijms232214046_

Round 1
Reviewer 1 Report
This manuscript presents a comprehensive overview of gibberellin-related genes involved in grain yield. However, confusing descriptions and obvious errors exist. I have some concerns and suggestions below.
Although the Abstract mentions the effectiveness of GA-related gene editing with CRISPR/Cas9 for crop development, there are only a few descriptions of genome editing in the text.
Table1
This is well summarized but needs some modifications. Some examples are shown below.
Several gene abbreviations are common. It would be better to indicate the abbreviation as well.
The DELLA protein in rice is SLENDER RICE1, not SLENDER1.
GA20ox2 and Semi-dwarf1 in rice are described as different genes even though they are the same gene. Furthermore, it is not appropriate that 121 is cited in GA20ox2.
GA20ox4 is reported as PANICLE RACHIS LENGTH 5 (PRL5), which is involved in rice panicle traits. This should also be mentioned.
Line 156, GID1 is a nuclear receptor.
(Murase et al., 2008, Nature, DOI: 10.1038/nature07519)
Author Response
This manuscript presents a comprehensive overview of gibberellin-related genes involved in grain yield. However, confusing descriptions and obvious errors exist. I have some concerns and suggestions below.
[Author]: We have changed the manuscript following your comments and suggestions. The revised parts are highlighted in red font in the new version of the manuscript.
Although the Abstract mentions the effectiveness of GA-related gene editing with CRISPR/Cas9 for crop development, there are only a few descriptions of genome editing in the text.
[Author]: CRISPR/Cas9 is an emerging technology in crop development studies. Thus, the available and relevant studies are limited to review, and we reviewed the available literature on the topic. To address this comment, we added the following section to the new version in the manuscript on page, lines: 404-414.
Table1
This is well summarized but needs some modifications. Some examples are shown below.
Several gene abbreviations are common. It would be better to indicate the abbreviation as well.
[Author]: Yes. We added the necessary abbreviations.
The DELLA protein in rice is SLENDER RICE1, not SLENDER1.
[Author]: Changed as suggested.
GA20ox2 and Semi-dwarf1 in rice are described as different genes even though they are the same gene. Furthermore, it is not appropriate that 121 is cited in GA20ox2.
[Author]: Changed as suggested.
GA20ox4 is reported as PANICLE RACHIS LENGTH 5 (PRL5), which is involved in rice panicle traits. This should also be mentioned.
[Author]: Changed as suggested.
Line 156, GID1 is a nuclear receptor.
(Murase et al., 2008, Nature, DOI: 10.1038/nature07519)
[Author]: Changed as suggested.
Reviewer 2 Report
The author systematically summarized the pathways involved in gibberellin biosynthesis and cell signal transduction, and comprehensively summarized the role of gibberellin in crop development . A few suggestions are for reference only.
1. In the abstract, the author mentioned the application of gene editing technology(Crispr/cas9) in the future GA-related gene regulation of crop growth, but it was not discussed in the article, more details should be described based on previous studies. please add.
2. The references are too old, and relevant literatures on gibberellin research in recent years are added. For example, He J., Chen Q., Xin P., et al. CYP72A enzymes catalyse 13-hydrolyzation of gibberellins. Nature Plants, 2019, 5: 1057-1065.
Liu, H., Guo, S., Lu, M. et al. Biosynthesis of DHGA12 and its roles in Arabidopsis seedling establishment. Nat Commun 10, 1768 (2019). https://doi.org/10.1038/s41467-019-09467-5
3. Figure 1 should be re-organized. 13-H GA, 13-OH GA should be marked.
4. The description of “Function” is not accurate enough, which should be described as changing the specific way of gibberellin and finally resulted in the corresponding phenotype in Table 1 .
5. There are also a lot of research reports on the upstream regulation pathway of gibberellin biosynthesis, and it is suggested to add corresponding content, such as wox transcription factor, bhlh transcription factor, etc.
Author Response
The author systematically summarized the pathways involved in gibberellin biosynthesis and cell signal transduction, and comprehensively summarized the role of gibberellin in crop development . A few suggestions are for reference only.
The author systematically summarized the pathways involved in gibberellin biosynthesis and cell signal transduction, and comprehensively summarized the role of gibberellin in crop development . A few suggestions are for reference only.
[Author]: We have changed the manuscript following your comments and suggestions. The revised parts are highlighted in red font in the new version of the manuscript.
- In the abstract, the author mentioned the application of gene editing technology(Crispr/cas9) in the future GA-related gene regulation of crop growth, but it was not discussed in the article, more details should be described based on previous studies. please add.
[Author]: CRISPR/Cas9 is an emerging technology in crop development studies. Thus, the available and relevant studies are limited to review, and we reviewed the available literature on the topic. To address this comment, we added the following section to the new version of the manuscript on page, lines: 404-414.
- The references are too old, and relevant literature on gibberellin research in recent years are added. For example, He J., Chen Q., Xin P., et al. CYP72A enzymes catalyse 13-hydrolyzation of gibberellins. Nature Plants, 2019, 5: 1057-1065.
Liu, H., Guo, S., Lu, M. et al. Biosynthesis of DHGA12 and its roles in Arabidopsis seedling establishment. Nat Commun 10, 1768 (2019). https://doi.org/10.1038/s41467-019-09467-5
[Author]:More recent references have been added.
- Figure 1 should be re-organized. 13-H GA, 13-OH GA should be marked.
[Author]: Changed as suggested.
- The description of “Function” is not accurate enough, which should be described as changing the specific way of gibberellin and finally resulted in the corresponding phenotype in Table 1 .
[Author]: Changed as suggested.
- There are also a lot of research reports on the upstream regulation pathway of gibberellin biosynthesis, and it is suggested to add corresponding content, such as wox transcription factor, bhlh transcription factor, etc.
[Author]: Changed as suggested.
Reviewer 3 Report
Reviewer’s comments
The manuscript described GA, its biosynthesis, interaction with other hormones and its applications. The manuscript is well structured and well written. I think the manuscript is suitable for publications in IJMS as present form.
However, only minor typo errors present in the MS. Please consider the following point
1. Page 2 line 51: The 50 barley Sdw1/denso contains gibberellin 20-oxidase2 (HvGA20ox), the orthologous of OsGA20ox2, and REDUCED HEIGHT (TaRHT) with a DELLA protein in wheat shows a similar phenotype to the rice sd1-1 [8, 9, 10].
Ø Did the authors mean mutation?
Ø “The 50 barley Sdw1/denso contains gibberellin 20-oxidase2 (HvGA20ox) mutation, the orthologous of OsGA20ox2, and REDUCED HEIGHT (TaRHT) with a DELLA protein in wheat shows a similar phenotype to the rice sd1-1 [8, 9, 10].”
2. Page 3 line 105-106: Studies show that rice only contains a single KAO-like gene, whereas KAO is encoded redundantly by two genes in Arabidopsis, Pisum sativum and Helianthus spec [28, 29].
Ø Please check the specie name of Helianthus
3. Page 4 line 136: Other mechanisms for GA inactivation mainly include epoxidation via GA methyl transferase 1 (GAMT1) and (GAMT2).
Ø Please as full name GA methyl transferase 1 (GAMT1) and GA methyl transferase 2 (GAMT2)
4. Page 4 line 139: change carboxy to carboxyl
5. Figure 2: Please add description for FPF1
6. Page 9 line 287: Add full name for PAC
Author Response
The manuscript described GA, its biosynthesis, interaction with other hormones and its applications. The manuscript is well structured and well written. I think the manuscript is suitable for publications in IJMS as present form.
However, only minor typo error present in the MS. Please consider the following point
[Author]: We have changed the manuscript following your comments and suggestions. The revised parts are highlighted in red font in the new version of the manuscript.
- Page 2 line 51: The 50 barley Sdw1/densocontains gibberellin 20-oxidase2 (HvGA20ox), the orthologous of OsGA20ox2, and REDUCED HEIGHT (TaRHT) with a DELLA protein in wheat shows a similar phenotype to the rice sd1-1 [8, 9, 10].
Did the authors mean mutation?
[Author]: Changed as suggested.
- Page 3 line 105-106: Studies show that rice only contains a single KAO-like gene, whereas KAO is encoded redundantly by two genes in Arabidopsis, Pisum sativum and Helianthus spec [28, 29].
Please check the specie name of Helianthus
[Author]: Changed as suggested.
- Page 4 line 136: Other mechanisms for GA inactivation mainly include epoxidation via GA methyl transferase 1 (GAMT1) and (GAMT2).
Please as full name GA methyl transferase 1 (GAMT1) and GA methyl transferase 2 (GAMT2)
[Author]: Changed as suggested.
- Page 4 line 139: change carboxy to carboxyl
[Author]: Changed as suggested.
- Figure 2: Please add description for FPF1
[Author]: Changed as suggested.
- Page 9 line 287: Add full name for PAC
[Author]: Changed as suggested.